# Developmental Biology and Identification of a Garden Pest, *Otiorhynchus* (*Podoropelmus*) *smreczynskii* Cmoluch, 1968 (Coleoptera, Curculionidae, Entiminae), with Comments on Its Origin and Distribution

**DOI:** 10.3390/insects14040360

**Published:** 2023-04-04

**Authors:** Rafał Gosik, Peter Sprick, Małgorzata Wrzesień, Agata Dzyr, Oliver Krstić, Ivo Toševski

**Affiliations:** 1Department of Zoology and Nature Protection, Institute of Biological Sciences, Maria Curie-Skłodowska University, Akademicka 19, 20-033 Lublin, Poland; 2Curculio-Institute e.V. (CURCI), Weckenstraße 15, 30451 Hannover, Germany; psprickcol@t-online.de; 3Department of Botany, Mycology and Ecology, Institute of Biological Sciences, Maria Curie-Skłodowska University, Akademicka 19, 20-033 Lublin, Poland; 4Independent Researcher, Kołłątaja 11/2, 24-100 Puławy, Poland; 5Institute for Plant Protection and Environment, Banatska 33, 11080 Belgrade, Serbia; 6CABI, Rue des Grillons 1, 2800 Delémont, Switzerland

**Keywords:** weevil, Curculionoidea, Entiminae, *Otiorhynchus smreczynskii*, *O. rotundus*, *COI*, morphometry, larva, pupa, adult stage, biology, distribution, origin

## Abstract

**Simple Summary:**

The taxonomic status of *O. smreczynskii* Cmol., 1968 as a valid species was finally confirmed on the basis of the results of molecular, morphological and anatomical studies. An urban species, with a distribution closely associated with human activities, this weevil is regarded as a pest of gardens and ornamental plants. The set of characters used for its identification is provided. The structure of the female genitalia turned out to be useful for differentiating the sister species *O. smreczynskii* and *O. rotundus*. The entire life cycle, morphology of immatures and phenology of *O. smreczynskii* were examined in detail both in the laboratory and in the field. Host plants are listed and the unique feeding signs of *O. smreczynskii* are illustrated. The revised distributions of *O. smreczynskii* and *O. rotundus* show that the bisexual *O. rotundus* is being displaced by the parthenogenetic *O. smreczynskii*. The hypothetical origin of *O. smreczynskii* given here is based on the natural distribution of its host plants.

**Abstract:**

The mature larva and pupa of *Otiorhynchus smreczynskii* are described and illustrated with complete chaetotaxy for the first time. Five larval instars and the factors for larval growth are determined, and the larval development of this species is comprehensively described. In order to confirm species affiliation, selected larvae were subjected to genetic analysis (mtCOI). Host plants and unique feeding signs of some Entiminae species are presented, and all available data on development are documented and interpreted. Additionally, the morphometry of 78 specimens (48 of *O. smreczynskii* and 30 of *O. rotundus*) was examined in order to verify the usefulness of morphological features in distinguishing both species. The female genitalia of both species are illustrated, described and compared with each other for the first time. Finally, the updated distribution of *O. smreczynskii* is given, and a possible origin of *O. smreczynskii* and *O. rotundus* is proposed.

## 1. Introduction

*O. smreczynskii* was described as a *species novum* on the basis of two specimens (females) collected in Lublin in 1965. The first description of *O. smreczynskii* [1] contained one picture of a beetle (general view from the dorsal side) and one drawing of the *spiculum ventrale*, but no drawings of other elements of the reproductive apparatus. This work was a modest approach, based mainly on features found in many members of the subgenera *Melasemnus* (Reitter, 1912) and *Podoropelmus* (Reitter, 1912). It did not contain a comparative key for distinguishing the newly described species from its well-known sister species *Otiorhynchus rotundus* (Marseul, 1872) (syn. *O. rotundatus*, Siebold, 1847). This could have been due to the placement of the two species in different groups: *O. smreczynskii* in *Melasemnus* and *O. rotundus* in *Proremus* (Reitter, 1912). Reitter [2,3] erected several species groups for species with a multiple-pointed pro-femoral tooth or with fine denticles on it. He distinguished *Podoropelmus* from *Melasemnus* and *Proremus* by the dilated and raised suture interval at the tip of the elytra and *Proremus* from *Melasemnus* by the row of fine bristles or long hairs along the midline of the intervals. At the same time, however, he limited this differentiation and allowed that *Melasemnus* might have a row of hairs on the elytral slope and parted hairs on the remaining interval, which would make unequivocal identification difficult. The species group status was raised to subgenus level in the catalogue of Löbl and Smetana [4], and the most recent change of subgenus assignment was made by Alonso-Zarazaga et al. [5].

Although the external morphology of adult *O. rotundus* and O. smreczynskii have been completely described, e.g., by Smreczyński [6], Dieckmann [7] and Korotyaev et al. [8], none of the characteristics listed can be regarded as unique to either of them. Moreover, the original description of the female genitalia of *O. smreczynskii* was incomplete, and hence insufficient for identifying the species. Consequently, the review by Cmoluch [9], focusing on the female genitalia of a number of *Otiorhynchus* species, did not contain any more information about the reproductive apparatus of *O. smreczynskii*, and, in the case of *O. rotundus*, did not describe either the *spiculum ventrale* or the *receptaculum seminis*. The identity of *O. smreczynskii* thus remained ambiguous for several years. Frieser [10], for example, gave the body size (without rostrum) as 5–10 mm (in fact it is mostly less than 5 mm, with occasional slightly larger exceptions). In consequence, the features stated as being diagnostic turned out to be doubtful, as a result of which some of the specimens could not be unequivocally identified as either *rotundus* or *smreczynskii*.

Therefore, as part of the international “Molecular Weevil Identification” project, specimens identified as *O. smreczynskii* (based on their origin from the site where this species was frequently observed for several decades) were used for a comparative analysis (based on the similarity of the *COI* gene) with some specimens from a site in Poland where *O. rotundus* was known to occur. That the Polish specimens in fact belonged to *O. smreczynskii* was overlooked, which resulted in the same barcoding result for *O. rotundus* and *O. smreczynskii*, and consequently, in the synonymization of the two species: *O. smreczynskii* = *O. rotundus* [11]. Moreover, in 2016, GenBank published the sequence of the gene encoding the cytochrome oxidase of the specimens, which, based on morphological features, was identified as *O. smreczynskii* and published as such. Thus, information regarding this synonymization was reported in the subsequent catalogue of Palaearctic beetles [5].

This paper presents a complete set of information on the phenology, larval morphology and development, host plant interactions, anatomy of certain adult structures and conspicuous morphological features that, with support from genetic data, allows *O. smreczynskii* to be unequivocally differentiated from *O. rotundus*.

## 2. Materials and Methods

### 2.1. Source of Materials

A detailed overview of the material investigated below is provided in Appendix A (materials used for the genetic studies) and Appendix A (materials used for the morphometric analysis).

Data on phenology, including larval development, were obtained by means of regular pitfall (71 exx) and sweep net catches (698 exx), by breeding and soil searches during the Soil-dwelling Weevils Project, and occasionally later on (see [12,13]). Together with unpublished results, this permits a few preliminary comments on the development of *O. smreczynskii* to be made.

Two attempts were made to obtain larvae or pupae from the roots of breeding plants in the climate chamber of the JKI (Julius Kühn-Institut) in Braunschweig. The first attempt, was carried out in three small flowerpots with *Ligustrum vulgare*, *L. ovalifolium* and *Syringa vulgaris*, started on 27 April 2012 with 55 specimens, but yielded only 2 mature larvae (in August and November). The second attempt, conducted in 2015 and 2016 using two larger flowerpots containing *Ligustrum ovalifolium* and *Euonymus fortunei*, was started on 31 July 2015 with 12 specimens and yielded a sufficient number of larvae between December 2015 and June 2016. Larvae were observed in the *Ligustrum* flowerpot, some of which were removed for description, but others disappeared, and there was no way of knowing whether they had died, were overlooked or had metamorphosed into adult weevils. On 3 February 2016 and 5 April 2016, 5 and 17 mature larvae were counted in the *L. ovalifolium* flowerpot, respectively. The smaller number on 3 February 2016 resulted from a very careful search but without a meticulous examination of the soil. One pot, in which several larvae were placed, was taken to Hannover for routine monitoring on 24 April 2016. Six larvae had been obtained by that date, as well as two larvae on 14 June 2016, two pupae on 24 April and another pupa on 3 May 2016. Another two pupae from 2011 had been bred by Thorsten Ufer (at the Ellerhoop horticultural centre), who tested the susceptibility of *O. smreczynskii* larvae to entomopathogenic nematodes (Figure 1).

In June 2017, field-collecting, which had earlier failed to yield any larvae, was successful; a few larvae were taken from a newly established population in the plant garden of the JKI with *Ligustrum vulgare*, *L. ovalifolium* and *Syringa vulgaris*. Additionally, one pupa was found in the field together with the larvae. Breeding in the climate chamber of the JKI was carried out under long-day conditions (light 16 hours, temperature 18 °C).

### 2.2. Molecular Study

Based on morphology, specimens were selected at random from the entire material identified by Rafał Gosik and Peter Sprick and subjected to genetic analysis for confirmation of identity by comparing sequences of the barcoding region of the mitochondrial cytochrome oxidase subunit I gene (mt*COI*) to the referent sequence of *O. smreczynskii* deposited in GenBank (KU910973). Individual weevils were punctured between the 2nd and 3rd thoracic sternites, and total DNA was extracted from the whole specimen using the QIAGEN DNeasy^®^ Blood and Tissue Kit (Qiagen, Hilden, Germany) according to the manufacturer’s instructions. The barcoding region of the mitochondrial cytochrome oxidase subunit I gene (mt*COI*) was amplified using the primer pair LCO1490hem (5′–TTTCAACTAAYCATAARGATATYGG–3′) and HCO2198hem (5′–TAAACYTCDGGATGBCCAAARAATCA–3′) [14]. Amplifications were performed in a 20 μL final volume containing FastGene Reaction Buffer A with 1.5 mM MgCl_2_ (1×), an additional 2.25 mM of MgCl_2_, 0.6 mM of each dNTP, 0.5 μM of each primer, 1 U of FastGene Taq DNA polymerase (Nippon Genetics, Tokyo, Japan) and 1 μL of DNA extract. PCR cycles were carried out in a Mastercycler EP gradient S (Eppendorf, Hamburg, Germany), applying the following thermal steps: 95 °C for 5 min (initial denaturation), 35 cycles at 94 °C for 1 min, 48 °C for 1 min (annealing), 72 °C for 1.5 min and a final extension at 72 °C for 7 min. The amplified products of the mt*COI* gene were sequenced in both directions on an ABI Prism 3700 automated sequencer using the commercial services of Macrogen Europe (Amsterdam, The Netherlands). The taxonomic identity of the larvae and pupae was confirmed by comparing their sequences with that of adult specimens. Pairwise distances using the p-distance model were analysed using MEGA5 software (software version 5.2.2) [15]. The mt*COI* sequences obtained were deposited in the NCBI GenBank database under the following accession numbers: MZ951149-MZ951156).

The haplotype network of mt*COI* was inferred using the statistical parsimony method [16], as implemented in the software PopART version 1.7 (http://popart.otago.ac.nz, (accessed on 28 December 2022), with a confidence limit of 95%.

### 2.3. Morphology and Larval Instar Identification

The immature stages were observed and measured under a light microscope with calibrated ocular lenses (Olympus SZ60 (Olympus Corporation, Tokyo, Japan). Slide preparation basically followed May [17]. The larvae selected for study under the microscope were cleared in 10% potassium hydroxide (KOH), then rinsed in distilled water and dissected. After clearing, the head, mouthparts and body (thoracic and abdominal segments) were separated and mounted on permanent microscope slides in Faure–Berlese fluid (50 g gum arabic and 45 g chloral hydrate dissolved in 80 g of distilled water and 60 cm^3^ of glycerol) [18].

The following abbreviations are used for the larvae: body length (BL), body width (BW) (in abdominal segment 2) and head capsule width (HW); and for the pupae: body length (BL), body width (BW) (at the level of the mid leg pair) and head width (HW). The drawings and outlines were made using a drawing tube (MNR-1) installed on a stereomicroscope (PZO SK 14) and processed by computer software (Corel Photo—Paint X7, Corel Draw X7). The photographs were taken using an Olympus BX63 (Olympus Corporation, Tokyo, Japan) microscope and processed with Olympus cellSens Dimension software (version 1.18). The specimens selected for SEM (scanning electron microscope) imaging were gold-plated. TESCAN Vega 3 SEM was used to examine selected structures.

The general terminology and chaetotaxy follow May [17] and Marvaldi [19], while the antennae terminology follows Zacharuk [20]. The general terminology of the female genitalia follows Wanat [21], Franz [22] and Li and Liang [23].

As the data were not normally distributed, a non-parametric Mann–Whitney U test was applied to determine the differences in length of elytra, width of elytra, length of pronotum, width of pronotum, length of head, width of head, length of tibia and total length of funiculus and clava between the two species *O. smreczynskii* and *O. rotundus*. The level of statistical significance of differences between the medians for all analyses was *p* < 0.05. The statistical tests were performed with STATISTICA 13.0 software (StatSoft, Inc., Kraków, Poland).

To identify the larval instars, we used the method of Leibee et al. [24], developed by Sprick and Gosik [25]. In soil-dwelling weevils it is possible to ascertain the number of larval instars if the head widths of the first and last larval instars of a given species are known.

## 3. Results

### 3.1. Molecular Study

The final alignment of the mt*COI* sequences consisted of 645 bp, with a total of 28 (4.3%) polymorphic nucleotides, all parsimony informative. A very low level of variability was detected when the mt*COI* gene in *Otiorhynchus* spp. was analysed. Analysis of 55 *Otiorhynchus smreczynskii* specimens, spanning 13 different populations (Figure 2), revealed the presence of only one haplotype (smr1), whereas in 12 sequenced specimens of *O. rotundus* from two populations, two haplotypes were detected, namely, rot1 and rot2 (Figure 3). A single haplotype detected within the *O*. *smreczynskii* populations suggests an asexual, i.e., parthenogenetic, mode of reproduction. However, the low level of variability in *O. rotundus* specimens which presumably reproduce sexually is surprising.

The registered p-distance between the two haplotypes of *O. rotundus* was 0.2%, while that between *O. rotundus* and *O. smreczynskii* was 4.3% (Table 1).

### 3.2. Description of Immature Stages of Otiorhynchus smreczynskii

#### 3.2.1. Larva (Figure 4A–F)

First instar: head width: 0.250^4^, 0.267^1^, 0.270^2^Medium instars: head width: 0.730^1^, 0.700^1^Mature larva: head width: 0.960^1^, 1.050^6^, 1.13^4^, 1.136^1^Body length: 6.0–6.5 mmBody width in the widest area (third abdominal segment): 1.98–2.20 mmHead width of mature larvae: 0.96–1.13 mm

Body (Figure 4A): slender, curved, rounded in cross section. Prothorax smaller than mesothorax; metathorax almost as wide as mesothorax. Abdominal segments 1–5 of the same length; 6–9 tapering gradually to the terminal parts of the body; 10 reduced to four anal lobes of equal size. Chaetotaxy well developed, setae capilliform, variable in length, light yellowish. Each side of prothorax with 12 *prns* of unequal length (10 placed on premental sclerite, next 2 close to spiracle), 2 *ps* and 1 *eus*. Meso- and metathorax (Figure 4B,D) on each side with one moderately long *prs*, five *pds* (first and third very long, second, fourth and fifth moderately so), one moderately long *as*, one moderately long *eps*, one *ps* and one *eus*. Each pedal area of thoracic segments with six *pda*, variable in length. Abd. 1–8 (Figure 4C,E,F) on each side with one moderately long *prs*, five *pds* (first, third and fifth very long, second and fourth short), arranged along the posterior margin of each segment, one min and one very long *ss* (Abd. 8 with *ss_1_* only), two *eps* and one moderately long and one long *ps*, one *lsts* and two moderately long *eus*. Abd. 9 (Figure 4C,E,F) on each side with three *ds*, first long, second and third short, all located close to the posterior margin of the segment, one long and one very short *ps* and two moderately long *sts*. Each lateral anal lobe (Abd. 10) with a minute seta.

**Figure 4 insects-14-00360-f004:**
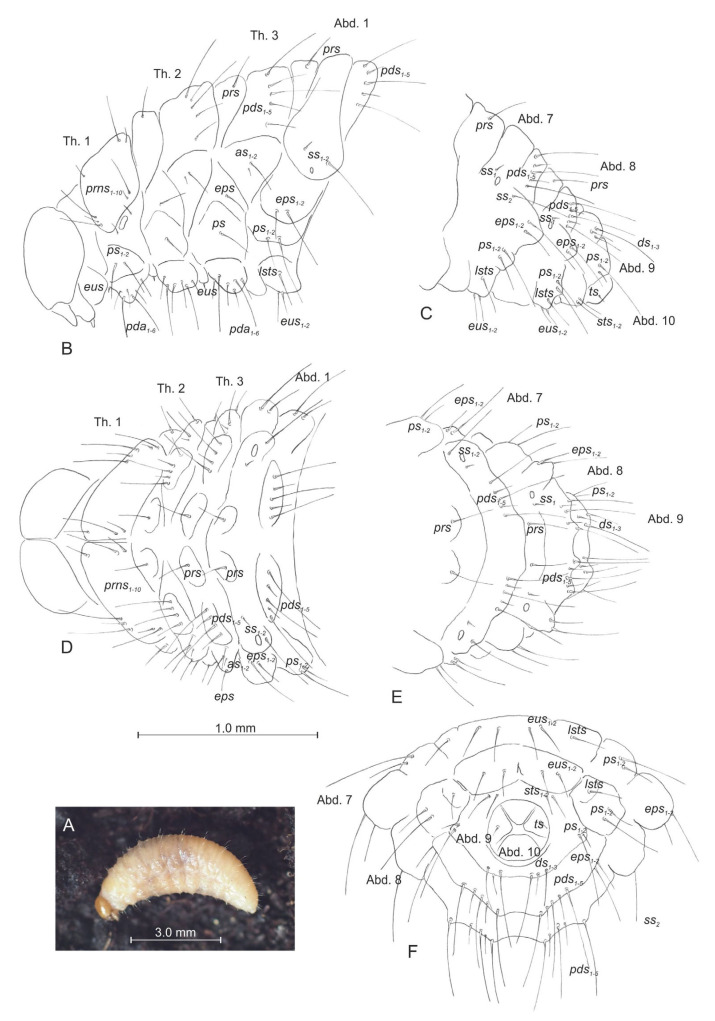
*Otiorhynchus smreczynskii,* mature larva, habitus and chaetotaxy. (**A**) Habitus; (**B**) lateral view of thoracic segments and first abdominal segment; (**C**) lateral view of abdominal segments 7–10; (**D**) dorsal view of thoracic segments and first abdominal segment; (**E**) dorsal view of abdominal segments 7–10; (**F**) ventral view of abdominal segments 7–10 (Th. 1–3—thoracic segments, Abd. 1–10—abdominal segments. Setae: *as*—alar, *ps*—pleural, *eps*—epipleural, *ds*—dorsal, *lsts*—laterosternal, *eus*—eusternal, *pda*—pedal, *pds*—postdorsal, *prns*—pronotal, *prs*—prodorsal, *ss*—spiracular, *sts*—sternal, *ts*—terminal*)*.

Head (Figure 5A,B): Light brown, suboval, frontal suture distinct, Y-shaped, endocarina absent. Setae on head capilliform. *Des_1,2,3,5_* very long, equal in length; *des_1_* and *des_2_* located in central part of epicranium, *des_3_* placed on frontal suture, *des_5_* located anterolaterally. Two minute setae, *des_4a_* and *des_4b_*, located between *des_3_* and *des_5_*. *Fs_4_* and *fs_5_* equal in length, *fs_4_* located anteromedially, *fs_5_* anterolaterally. *Les_1_* and *les_2_* half the length of *des_1_*. *Ves_1_* and *ves_2_* short, poorly developed. Postepicranial area with four very short *pes_1–4_*. A pair of small stemmata (st) located anterolaterally on each side of the head.

Antenna (Figure 6) located at end of frontal suture; antennal segment membranous with cushion-like Se, located medially and with five sensilla of different types: two sa and three sb.

Labrum (Figure 7A) semicircular; three pairs of *lrs* of various length, *lrs_1_* moderately long, *lrs_2_* long, *lrs_3_* short; *lrs_1_* and *lrs_2_* placed medially, *lrs_3_* anterolaterally.

Clypeus (Figure 7A) relatively wide, trapezium-shaped, anterior margin of clypeus straight; two pairs of *cls* of various length, *cls_1_* very short, *cls_2_* almost as long as *lrs_3_*, both located posteromedially; clss clearly visible, placed medially between *cls*. Epipharynx (Figure 7B) with three pairs of long, rod-shaped *als* of almost equal length; three pairs of *ams*: *ams_1_* and *ams_2_* rod-shaped, *ams_3_* capilliform; *ams_1_* very short, *ams_2_* and *ams_3_* half the length of *als*; two pairs of rod-shaped *mes*, first pair placed anteriorly, very close to *ams*, second pair medially, *mes_1_* distinctly longer than *mes_2_*. Anterior margin of epipharynx smooth, medial part serrate owing to presence of squama-like asperities surrounding labral rods. Labral rods long, strongly converging posteriorly.

Mandibles (Figure 8) curved, moderately narrow, with divided apex (teeth variable in length). A protruding additional tooth on cutting edge between apex and middle of mandible; both *mds* capilliform, variable in length.

Maxilla (Figure 9A–C) with one *stps* and two *pfs* of equal length; mala with seven *dms*: 1st–3rd capilliform, 4th–7th finger-like, unequal in size (Figure 9B), and four *vms* (Figure 9C); *vms* distinctly smaller than *dms*; *mbs* short. Maxillary palpi with two palpomeres, basal with short *mps*; distal palpomere apically with a group of sensilla, each palpomere with a pore. Basal palpomere wider than distal, both of almost equal length. Praelabium (Figure 9A) rounded with one long *prms*, located medially. Ligula with a pair of capilliform, moderately long *ligs*. Premental sclerite clearly visible, trident-shaped. Labial palpi two-segmented; apex of distal palpomere with some sensilla; each palpomere with a pore. Basal palpomere distinctly shorter than distal. Postlabium (Figure 9C) with three capilliform *pms*, the first pair located anteromedially, the other two pairs laterally; *pms_1_* and *pms_3_* short, half the length of *pms_2_*.

#### 3.2.2. Larval Instar Identification

For instar identification in *O. smreczynskii*, a Growth Factor (GF) of 1.43 (compared with 1.42 and 1.44; in brackets) leads to the best approximation of the three tested values (L_1_ with 0.258 mm): 0.369 mm (0.366/0.372) for L_2_, 0.528 mm (0.520/0.535) for L_3_, 0.754 mm (0.739/0.770) for L_4_ and 1.079 mm (1.049/1.109) for L_5_ larvae. The GF is very close to the GF values of 1.44 and 1.46 that yielded the best approximation(s) in *Tanymecus* species [26]. Hence, we can state that *O. smreczynskii* has five larval instars and that the unknown instar in Table 2 appears to belong to larval instar 4, i.e., the penultimate larval instar. Moreover, this is validated by the fact that the head widths of the pupa and of the adult instar are very close to the head width of the last larval instar.

#### 3.2.3. Pupa (Figure 10A–D)

Body length: 4.5–5.1 mm ♀; body width: 2.6–3.2 mm ♀; thorax width: 1.4–1.6 mm ♀. Head width: 0.97–1.00 mm ♀.

**Figure 10 insects-14-00360-f010:**
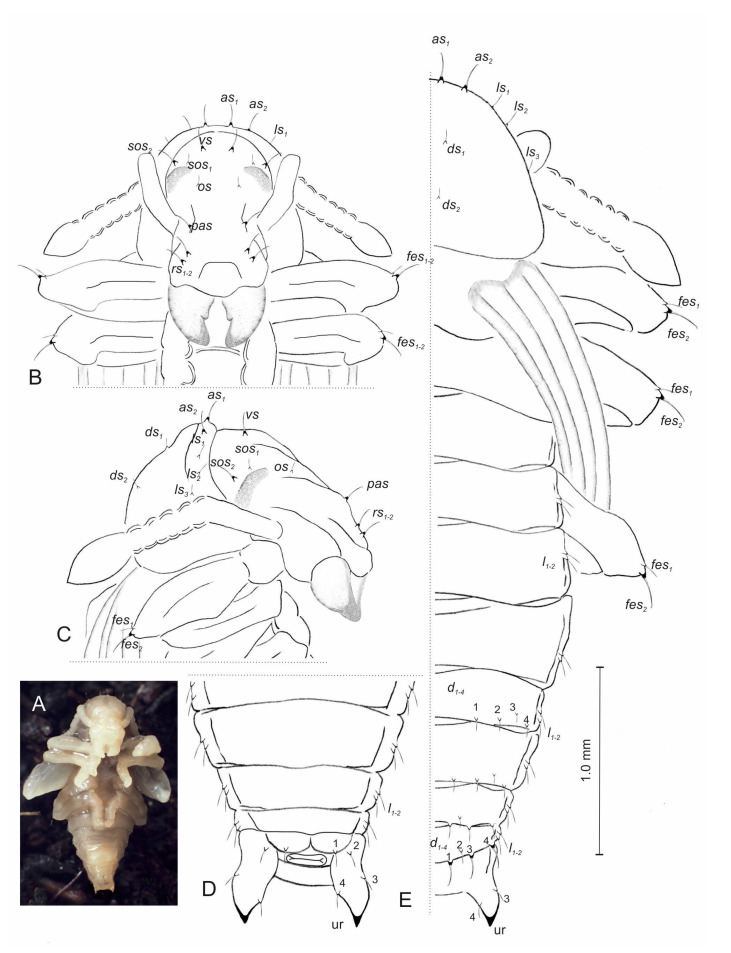
*Otiorhynchus smreczynskii,* pupa. (**A**) Habitus; (**B**) head, dorsal view; (**C**) head and pronotum, lateral view; (**D**) ventral view of last abdominal segments; (**E**) dorsal chaetotaxy (ur—urogomphus. Setae: *as*—apical, *d*—dorsal, *ds*—discal, *fes*—femoral, *l*, *ls*—lateral, *os*—orbital, *pas*—postantennal, *rs*—rostral, *sos*—superorbital, *vs*—vertical).

Body rather slender, slightly curved, whitish. Cuticle smooth. Rostrum short, 1.2 times as long as wide, extended beyond procoxae. Antennae moderately long. Pronotum almost 1.5 times as wide as long. Abdominal segments 1–3 of almost equal length, segments 4–7 tapering gradually, 8 semicircular, 9 distinctly smaller than previous segments. Urogomphi moderately long, conical, with sclerotized apical parts.

Chaetotaxy rather poorly developed, setae variable in length, capilliform, straight or slightly curved. Setae light yellowish, usually located on small protuberances. Head capsule and rostrum (Figure 10A) include one *vs*, two *sos*, one *os*, one *pas* and two *rs*. *Vs*, *sos_1_*, *pas* and *rs_1_*_–*2*_ moderately long, with curved apical parts. *Sos_2_* and *os* very short, straight. Pronotum (Figure 10B) with two *as*, three *ls* and two *ds*. *As_1–2_* long, with curved apical parts, remaining setae straight, moderately long or short. 

Meso- and metathorax without setae. Dorsal parts of abdominal segments 1–4 without setae and each of segments 5–8 with four pairs of setae; *d_1–4_* placed posteriorly in regular lines. Each of abdominal segments 1–8 with two *l_1–2_*. Dorsal setae of abdominal segment 8 distinctly longer than those of previous segments. Ventral parts of abdominal segments without setae (Figure 10C). Abdominal segment 9 with four pairs of setae: the first placed on gonothecae, the next on urogomphi. Each apex of femora with two *fes*; *fes_1_* moderately long, straight, *fes_2_* long, with curved apical part (Figure 10D).

### 3.3. Biology: Development and Phenology

The first overwintering adults became active already in the second half of April; egg-laying was observed in late May–early June and later. In the lab, batches of oval, white eggs, turning light brownish after a few days, were laid in groups even on the bare bases of the storage boxes, between layers of paper or in corners. In the field, a few mature (or premature) larvae were found in June as well as a single pupa under *Ligustrum vulgare* shrubs at the same time. In the climate chamber, when breeding was started on 31 July, larvae were obtained in December and from February to June in the following year; in April and May a few pupae were found as well. All this indicates that overwintering of larvae is probable when eggs are laid late in the season.

Data on adult phenology: In 2009, teneral adults, recognized mainly by their pale colouring or smooth cuticle, were found from between June and August 2009, with the main emergence in August (Figure 11); in 2011, immature adults were caught in small numbers on two days in April and again in higher numbers from the beginning of July and in August (Figure 12). Thus, the development of most specimens should take place between April/May and July/August. The few immature adults in April point to overwintering individuals, as low temperatures delay hardening of the cuticle throughout the winter. Immatures may have developed into the adult stage during the last warm days in September or the beginning of October, subsequent development then being delayed by unfavourable conditions (larval development was also delayed in the climate chamber in winter). In September and October. the numbers of adults were the lowest, but a high proportion of these were teneral. This was indicated by minimal climbing and moving activity above the soil surface, and, probably, that an unknown proportion remains as teneral adults inactive in the soil. It is assumed that teneral adults, in contrast to hardened specimens, still need to take up food when temperatures are still high enough. As a result, those specimens may be caught late in the season, when most others have already moved to their hibernation sites deep in the soil.

Breeding data from the climate chamber, starting at the end of April, show that early active beetles produced mature larvae in the same year, whereas breeding which started at the end of July produced mature larvae in December of the same year.

In 2012 the following sweep net data were obtained: 2 tenerals among 33 adults on 20 April 2012, 0 of 62 on 11 May 2012, 0 of 36 on 1 June 2012, 1 of 27 on 24 June 2012 and 10 of 18 on 6 July 2012 [13], thereby confirming the results from 2011. A small number of adults overwinter as tenerals and harden in May and June, whereas the first weevils that developed completely in 2012 appeared in the second half of June and were abundant in July. The emergence period of new adult beetles is prolonged until September and by single specimens even into October.

In the field, the main means of development of *O. smreczynskii* is probably an early egg-laying period and larval development in the same year; larval overwintering in the field is apparently less common than in the climate chamber, where larval development was strongly retarded during the winter months (diapause).

### 3.4. Comparison of External Morphology in Adults of Otiorhynchus smreczynskii and O. rotundus

The morphometric studies account for the variability of eight morphological features: a—length of elytra, b—width of elytra, c—length of pronotum, d—width of pronotum, e—length of head and rostrum, f—width of head, g—length of fore tibia, h—total length of funiculus and clava, i—length of the body (without rostrum) (Figure 13A). The range of variability of these features is given in Table 3 and Table 4. The overall results of the measurements are listed in Appendix A. We also compared the shapes of setae and scales.

Comparison of the average values showed that most of the measured features displayed statistically significant differences between the two species, but the large overlaps of all measurements may explain why distinguishing between these species on the basis of morphometric differences is not recommended (Table 4, Figure 14).

The distinction between *O. rotundus* and *O. smreczynskii* was hitherto based on the shape of the teeth on the femora and the number of spines on the inner surface of the pro-tibiae. Because these features are highly variable, especially in *O. rotundus*, they do not always yield unequivocal results.

Morphological details and body size of *O. rotundus* are extremely variable. The shape of the elytra varies from slightly elongate to almost rounded. While the body colour varies from light brown to dark brown, the colouration of the head, pronotum and elytra is always uniform. The pattern of scales is very variable: sometimes they form distinct bands on the elytra, sometimes the pattern is reduced to isolated spots, but it is always present. The teeth on the fore femora are undivided, conical, almost smooth (Figure 15A–I), or equipped with small lateral appendages. It is rarely split, but if it is, the split never extends to the base of the tooth. The spines on the inner part of the front tibia are small, hardly visible among the bristles. In addition, certain differentiation between the sexes is impossible without dissection of the genitalia.

### 3.5. Description of Female Genitalia of Otiorhynchus smreczynskii and O. rotundus

As no illustrated descriptions are available, we now describe and compare the female genitalia of *O. rotundus* and its sister species *O. smreczynskii*.

*O. smreczynskii*: tergite of abdominal segment 8 (Figure 16A) subtrapezoid, apical margin smooth; sternite rounded (Figure 16B), sparsely, regularly stippled, both parts densely covered with elongate, hair-like setae; *spiculum ventrale* (sv) (Figure 16C) 3.75 times longer than apical lobe, ensiform, straight.

*O. rotundus*: tergite of abdominal segment 8 (Figure 16D) ellipsoid; sternite (Figure 16E) with subtrapezoidal apical lobe (apl), sinuate at base, both parts moderately covered with hair-like setae, variable in size; *spiculum ventrale* (Figure 16F) four times longer than apical lobe (apl), ensiform, curved.

*O. smreczynskii*: spermatheca C-shaped (Figure 17A); ramus (ra) and collum (cl) not projected and well separated; corpus (co) moderate in size, cornu (cr) distinctly longer than corpus, curved, distinctly tapering to the top, apex (ap) acute; ovipositor (Figure 17B) with coxites (cx) elongate, cylindrical, densely stippled, densely covered with elongate setae, stylus (sy) cylindrical, short, with some variable setae on the apex.

*O. rotundus*: spermatheca C-shaped (Figure 17C); ramus (ra) and collum (cl) not projected and well separated; corpus (co) tumescent; cornu (cr) very elongated, only slightly tapering to the top; ovipositor (Figure 17D) with coxites (cx) sparsely stippled, sparsely covered with setae, stylus (sy) very short.

## 4. Discussion

### 4.1. Morphology and Species Identification

The first information on the morphological structure of *O. rotundus* larvae (as *O. rotundatus* Siebold, 1847) was published by Lengerken [27,28] and partially re-published by Scherf [29]. Even this very early approach to the description of a weevil larva indicates the existence of significant morphological differences between the developmental forms of *O. rotundus* and *O. smreczynskii*. The larva of *O. smreczynskii* has five *des* and six *pda*, and the pupa has two *sos*, three *ls* and two *ds* but no *sls*. The larva of *O. rotundus* has four *des* and five *pda*, and the pupa has one *sos*, one *ls*, one *ds* and one *sls.*

The chaetotaxy of the larvae and pupae of *O. smreczynskii* is very similar (next to *O. rotundus*) to that of the similar-sized species *O. crataegi* and *O. ovatus* (Linnaeus, 1758) [30].

Although all the measured morphological components are statistically significant, their usefulness for distinguishing between the two species may be limited in practice owing to the large morphological variation observed in both species (Table 3). The probable cause of the incorrect identification of some specimens could be that too much attention has been paid to features that are highly variable. Moreover, inaccuracies could also ensue from the different measuring techniques employed: Cmoluch [1] measured body length (=body size) including snout (=rostrum) length, whereas Smreczyński [6] and Dieckmann [7] did not include the snout. In addition, the holotype, described by Cmoluch [1], was an exceptionally large specimen, as long as 5.8 mm including the rostrum.

As already proposed by Korotyaev et al. [8], we confirm that the most valuable morphological trait permitting the unequivocal distinction between the two species is the shape of the scales. In *O. smreczynskii*, hair-like, sharply-pointed, smooth scales are dominant in the rear part of the integument, whereas in *O. rotundus*, they are wide, rounded and clearly crimped (Figure 18A–I). When in doubt (e.g., in dirty or worn specimens), the simplest and quickest method of distinguishing between the two species is to look for the presence or absence of the proximal tooth on the inner side of the pro-tibiae or to compare the scales on the heads. In *O. smreczynskii*, the space between the eyes is covered with uniform, elongate, hair-like scales, while in *O. rotundus* there are two types of scales, hair-like and wide, rounded with a metallic sheen (Figure 18A–I).

Other morphological features useful in distinguishing *O. smreczynskii* from *O. rotundus* include the shape of the dimples on the elytra: in *O. smreczynskii* the dimples have smooth edges, while in *O. rotundus* their edges are serrated. Moreover, some specimens of *O. smreczynskii* have a black head and a black pronotum but brown elytra, whereas the head, pronotum and elytra in *O. rotundus* are always of the same colour.

The greatest differences between the two species among the components analysed are in the structure of the *ovipositors* (Figure 17B,D). In *O. smreczynskii*, two-thirds of the surface of the valvifer, measured from the apex, are covered with relatively elongate setae, while in *O. rotundus*, the setae (medium or short) cover about one half of the area of the valvifer (measured from the tip). There are also differences between *O. rotundus* and *O. smreczynskii* in the shape of abdominal segment 8, the *spiculum ventrale* and abdominal sternite 8 (Figure 16). In *O. smreczynskii*, abdominal segment 8 is subtrapezoid and covered with elongate setae, the *spiculum ventrale* is straight and the apical lobe is rounded at the base, bearing apical setae of variable size. In *O. rotundus*, by contrast, abdominal segment 8 is ellipsoid, covered with setae, and medium-sized; the *spiculum ventrale* is curved, and the apical lobe is ensiform.

According to Kangas and Rutanen [31], the shapes of the *spiculum ventrale* and *receptaculum seminis*, despite the presence of distinct species characteristics, may also be subject to significant variability. Moreover, Korotyaev et al. [8] pointed out the morphological variability in the shape of the genitalia of *O. rotundus*.

### 4.2. Biology

Lengerken [27,28] stated that the bisexual species *O. rotundus* produced larvae and pupae in late July and in August. His observations were based on a few larvae, one pupa and a teneral adult. This may also be the period with the main emergence of new adults of *O. smreczynskii*, as confirmed by pitfall trap and regular sweep net catches, which yielded the highest proportion of teneral adults of both species in July and August (Figure 11 and Figure 12).

There is much information about the characteristic feeding niches of these species, their preference for Oleaceae and synanthropic environments in central Europe, their current west- and northward spread, especially of *O. smreczynskii*, and their strictly nocturnal activity (e.g., [12,27,28,32,33,34,35,36,37]).

Burkhardt [38] and Lengerken [32] (repeatedly) drew the unique feeding signs made by *O. rotundus*, especially on *Syringa vulgaris* L. leaves, but also on *Ligustrum vulgare* L., *Philadelphus coronarius* L., *Spiraea salicifolia* L. and *Cornus sericea* L. (the last-mentioned under the name of *C. stolonifera* Michx.). Burkhardt [38] emphasized that apart from *Syringa, Ligustrum* and *Symphoricarpos*, most shrubs grew in close proximity to heavily populated *Syringa* shrubs. All these observations were made in the former West Prussia, now the Gdańsk and Bydgoszcz regions of present-day Poland.

Schulze [39] and Richter [33], still using the appellation of *O. rotundatus*, subsequently revised by Dieckmann [7], Sprick [40] and Cmoluch and Czarniawski [41] to *O. smreczynskii*, were motivated to illustrate the feeding signs of this species on *Syringa* and *Ligustrum* leaves, which, though equally characteristic, are indistinguishable on these two species. The map showing the distribution of *O. smreczynskii* in the city of Hannover was, after some initial verification, based only on these signs, which were recorded at 16 of 126 sites on the leaf margins of *Syringa vulgaris* bushes and mainly on the basal leaves of small trees [40]. In 1989, the species was not as common as it is today, when it can be found on nearly every *Syringa* bush in the entire city.

Although these feeding signs (Figure 19A–H) are very similar to those of *O. rotundus*, they are in fact so characteristic that the presence of one of these species can be predicted from them with a high probability (>99%) if present on Oleaceae, and especially on *Syringa vulgaris*, *S. josikaea* Jacq. ex Rchb., *Ligustrum vulgare* or *L. ovalifolium* Hassk. in central Europe. Both species exhibit similar food preferences and occur in the same (anthropogenic) habitats in large parts of their ranges. Other host plants of *O. smreczynskii* include *Lonicera pileata* Oliver, *Viburnum* × *bodnantense* Aberc. ex Stearn, *Symphoricarpos albus* (L.), *Symphoricarpos* × *chenaultii* Rehder and *Spiraea hypericifolia* L. [13]. To these, Smreczyński [6] and Dieckmann [7] added *Cornus sanguinea* L., *Laburnum anagyroides* Medik. (as *Laburnum vulgare* Bercht. & Presl), *Robinia pseudoacacia* L., *Crataegus crus-galli* L., *Ribes aureum* Pursh and *R. sanguineum* Pursh. Both species are sometimes found on bushes other than Oleaceae if growing in close proximity to the latter, as reported by Burkhardt [37] for *O. rotundus* on *Cornus alba* L., *Cornus sanguinea* L., *Prunus serotina* Ehrh., *Ribes aureum* and *Rosa canina* L., and by Sprick [13] for *O. smreczynskii* on a *Rosa chinensis* Jacq. hybrid. Several of the plants listed by Dieckmann [7], among them *Cotoneaster* spec., *Forsythia* × *intermedia* Zabel and *Symphoricarpos* × *chenaultii*, may also belong to this category. However, *O. smreczynskii* has also been found on some Caprifoliaceae (*Lonicera pileata*, *Symphoricarpos albus* and *Viburnum* × *bodnantense*) and a few Rosaceae (mainly *Spiraea*), regardless of Oleaceae growing nearby, which appears to confirm that larval development can take place on some more lignified plant species growing some distance away from sites with Oleaceae bushes.

The biology, habitat and host plant preferences, as well as the strictly nocturnal activity of *O. rotundus* and *O. smreczynskii* are so similar that one might think they belonged to a single species.

In view of ongoing introductions, however, it is well to bear in mind that further southeast European species of the subgenus *Podoropelmus* and the related subgenus *Melasemnus* can produce similar feeding signs, e.g., *O. albidus* Stierlin, 1861, found in St. Petersburg, and *O. ukrainicus* Korotyaev, 1984, closely related to *O. smreczynskii*, even if present on Oleaceae (see [8,41,42,43]).

Sautkin and Meleshko [44] are mistaken if they consider that the feeding signs of *Dodecastichus inflatus* (Gyllenhal, 1834), *Otiorhynchus meridionalis* Gyllenhal, 1834 and *O. hungaricus* Germar, 1824 can be confused with those made by *O. smreczynskii* or *O. rotundus*, at least if present in larger numbers. The feeding habits of all these species except *O. hungaricus* were studied during the Soil-dwelling Weevils Project [45]: feeding signs were observed in the field, reproduced in the lab and illustrated by several photographs [12]. They all differ clearly from those made by *O. smreczynskii* and the nearly identical signs made by *O. rotundus:* the edges of the leaves look as if they have been punched out very narrowly (Figure 19A–H). *O. hungaricus*, not present in Germany, is replaced there by the closely related, similar-looking *O. clavipes* (Bonsdorff, 1785) and *O. fagi* Gyllenhal, 1834, whose very different, larger feeding signs were studied in the city of Hannover and in the nearby foothill regions in the south on leaves of *Fraxinus excelsior* L., *Fraxinus ornus* L.*, Euonymus fortunei* (Turcz.) Hand.-Mazz., *E. japonicus* Thunb. and *Syringa vulgaris* (Figure 19I–N). To assume that feeding signs of a sawfly, *Macrophya punctumalbum* (Linnaeus, 1767) (Hymenoptera, Tenthredinidae), can be confused with those of *O. smreczynskii* or *O. rotundus* seems very strange [44].

The paper by Korotyaev and Andreeva [36] largely confirms our observations about the host plants of both species as well as the fact that the feeding signs are almost instantly recognizable. These authors mention a further *Otiorhynchus* species, *O.* (*Melasemnus*) *lederi* Stierlin, 1876, as producing very similar feeding signs on *Syringa, Ribes* and *Fragaria* in eastern Turkey.

### 4.3. Origin and Distribution

*O. rotundus* has been reported in Lithuania, Poland, Moldova, Ukraine, Hungary, Slovakia, Austria [46], Belarus and the European part of Russia [4,47], including the Kaliningrad region [48]. The origin of this species, described from Poland by Siebold [49] under the name of *rotundatus* from the district of Danzig (Gdańsk) called Heubude (presently known as Stogi), should be Ukraine and perhaps the adjacent parts of southern Russia. Old records given by Reitter [3] and cited by Burkhardt [38] refer to “Podolia” and “eastern Galicia”, historical provinces situated mainly or completely in the west and south of present-day Ukraine, and to “Russia” (without any further locality details). The historical distribution of *O. rotundus* may have included Moldova and south-eastern Poland, as the western borderlands of eastern Galicia and the southernmost extent of Podolia could have been part of these countries. Yunakov et al. [50] reported that *O. rotundus* inhabits natural habitats in the Kharkiv region of eastern Ukraine, such as lowland deciduous forests along large rivers, which is the only indication of a non-anthropogenic habitat for this species. The distribution of *O. smreczynskii* ranges from eastern to central, northern and eastern parts of western Europe; it also occurs in southwestern Siberia [51], very probably as an introduction (Figure 20). It has been listed in a number of European countries, including Belarus, Denmark, Germany, Latvia, the Netherlands, Moldova, Russia, Switzerland and Sweden [8,34,36,37,40,44,48,52], Egorov (personal comm. from 02.07.2021), thus showing a significantly wider distribution than *O. rotundus*. However, the data on the occurrence of this species in Estonia and Lithuania [34] are most likely incorrect (the photos in the paper show evidently male *O. rotundus*).

According to Heijerman and Burgers [36] and Cmoluch (unpublished data), the spread of *O. smreczynskii* is purely anthropogenic. Cmoluch (unpublished data) and Dieckmann [7] presumed this species to have originated in Belarus, and the information was reproduced by many other authors. This concept has since been revised by Sautkin and Meleshko [44] and Korotyaev and Andreeva [37], the former authors having recorded this species in Belarus for the first time in June 2015. The origin of *O. smreczynskii* is still not well known. However, there is a record of a single specimen in a natural habitat in the Kharkiv region of Ukraine [50], and Korotyaev and Andreeva [37] list a few further specimens (without giving details of sex), from nature reserves and forest areas in the Russian provinces (oblasts) of Belgorod and Rostov. These records provide grounds for hypothesizing that eastern Ukraine and the adjacent regions of Russia could belong to the regions of origin of this species, thereby confirming at least in part the listed distribution data from Magnano and Alonso-Zarazaga in Löbl and Smetana [4]. No possible host plants in natural biotopes in Ukraine or Russia are known for either species. Owing to the very great similarity in host plant use, behaviour, habitus and regions of origin, one could also hypothesize that *O. smreczynskii* has evolved directly from *O. rotundus*, all the more so as a bisexual population of *O. smreczynskii* has never yet been found.

Under laboratory conditions, development and egg deposition were observed several times. Given that all the anatomical structures required for copulation and sperm storage have been fully preserved in females of *O. smreczynskii*, this supports the hypothesis that parthenogenesis is an evolutionarily young phenomenon that has subsequently evolved in numerous animal and plant groups [53]. In addition, it has been proposed that asexual reproduction in insects has been favoured in groups with limited or no dispersal ability [54]. As a flightless soil-dwelling weevil, *O. smreczynskii* certainly falls within this category, especially when over 60 species belonging to the genus *Otiorhynchus* have been found to be parthenogenetic [55]. Furthermore, parthenogenesis in flightless soil-dwelling weevils is apparently linked with their presence in Ice Age environments and the prevention of migrations by large mountain barriers [56]. Since asexuality and polyploidy are tightly linked, further research is needed regarding the number of chromosome sets in *O. smreczynskii.*

In *O. smreczynskii*, the area of origin could be delineated by the presence of males, as in other *Otiorhynchus* species with mainly parthenogenetic populations. Bisexual populations of many species usually inhabit small or very small areas, and several have been able to enlarge their ranges by parthenogenesis [56]. Mazur [57] demonstrated this for *O. coarctatus* Stierlin, 1861, *O. crataegi* Germar, 1824, *O. raucus* (Fabricius, 1777) and *O. rhilensis* Stierlin, 1888, and [56] for *O. scaber* (Linnaeus, 1758), now named *O. carinatopunctatus* (Retzius, 1783). Additionally, it is suggested that asexuality and specifically polyploidy in insects enhance colonizing success in comparison with sexual populations or species [55].

Without doubt, the expansion of *O. smreczynskii* is strictly anthropogenic [8,36,41], Egorov (pers. comm. from 02.07.2021). However, the beetles’ own dispersal abilities are rather small. Field observations indicate that *O. smreczynskii* individuals usually remain in the immediate vicinity of their reproductive sites. The beetles usually move once a day vertically from the soil to the host plant and back, but they are also capable of moving from garden to garden and to spread within towns and villages over the years (in 1989 the species was rather rare in the city of Hannover, but now it is present nearly everywhere). The usually high abundance of *O. smreczynskii* favours this kind of spreading behaviour. Despite the presence of suitable host plants, it only rarely reaches habitats outside built-up areas.

The plant communities of Lublin were meticulously explored by Cmoluch and other researchers as part of investigations into the entomofauna of the xerothermic communities of the Lublin Upland [58]. Cmoluch (pers. comm.) doubts therefore whether the population of *O. smreczynskii* could have existed in Lublin before 1965. The mass appearance of this beetle in subsequent years must have been associated with the large-scale planting of roadside privet hedges. This leads to the inference that it must have been overlooked for several years prior to those plantings. Today, however, it is impossible to establish where the seedlings used for planting came from.

The species pair *O. rotundus* and *O. smreczynskii* is a perfect example of two species with a similar origin (east to southeast Europe), living in the same habitat, utilizing the same host plants and with a similar body size, demonstrating that the parthenogenetic species has been more successful in spreading and enlarging its distribution, a process that has intensified considerably during the past 60 years as a result of human agencies, when introductions and passive transport with plant material have strongly increased: for a new population to form, just one single individual suffices.

## 5. Conclusions

In August 2022, the second author checked some sites in Küstrin-Kietz and nearby, given by Burkhardt [8], to look for *O. rotundus*. However, at all sites, three in Küstrin-Kietz and two in Neubleyen, only *Otiorhynchus smreczynskii* was present. The data of Burkhardt about the presence of *O. rotundus* in the area of Küstrin (Golzow, where *O. rotundus* may have been replaced by *O. smreczynskii* in the meantime) are of interest in view of Dieckmann’s statement that before 1946 no specimen of *O. smreczynskii* was present in any collection from the former GDR [7], in this way obliquely confirming the historical presence of *O. rotundus* in eastern Brandenburg in 1917. Only 90 years later, first in 2007, the species was re-introduced to Germany (Thuringia) [59]. This corresponds with data from Poland, where there are no records of *O. smreczynskii* from before 1965 [58].

## Figures and Tables

**Figure 1 insects-14-00360-f001:**
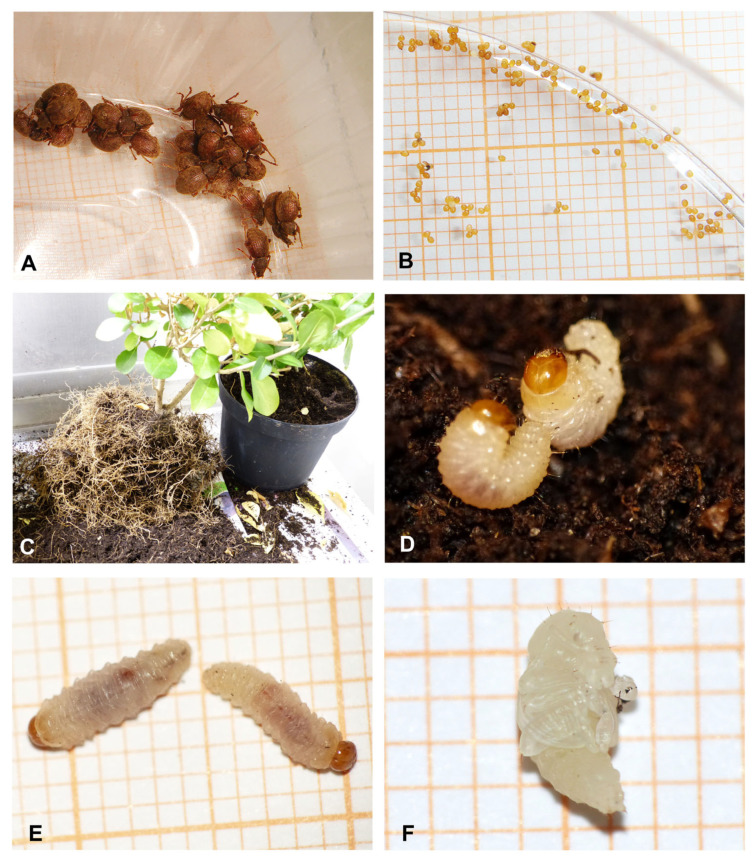
*Otiorhynchus smreczynskii*, breeding under laboratory conditions. (**A**) A group of adults collected for breeding, clustering undisturbed in the boxes, (**B**) eggs, (**C**) searching for immatures in a flowerpot with *Ligustrum ovalifolium*, (**D**) larvae collected from that flowerpot; (**E**) larvae on diagram paper, (**F**) pupa on diagram paper (phot. P. Sprick).

**Figure 2 insects-14-00360-f002:**
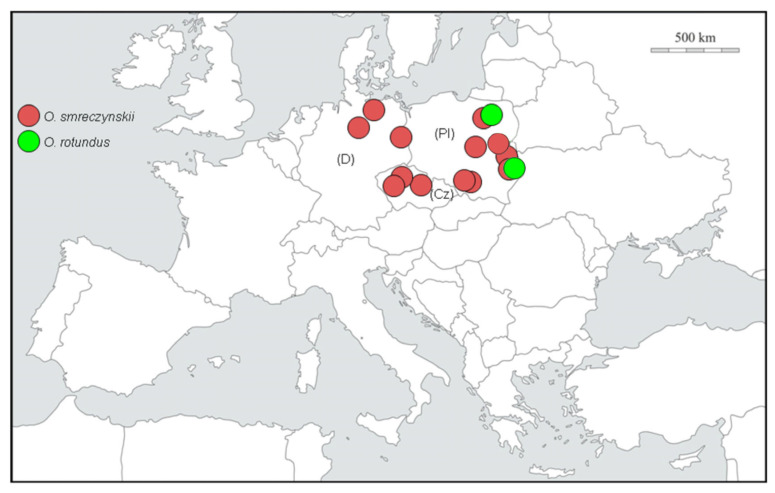
Sampling sites for *Otiorhynchus rotundus* and *O. smreczynskii* weevils in Poland (Pl), Germany (D) and the Czech Republic (Cz) used for genetic analysis. Map from d-maps.com (https://d-maps.com/carte.php?num_car=2232&lang=en, accessed on 20 January 2023).

**Figure 3 insects-14-00360-f003:**
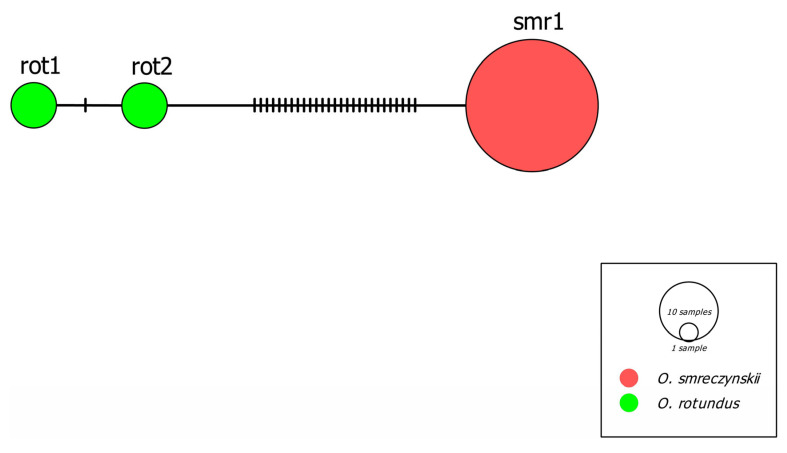
Statistical parsimony network constructed using PopArt obtained from cytochrome oxidase subunit I (mt*COI*) of *Otiorhynchus rotundus* and *O. smreczynskii*. The colours correspond to the species’ identities. The vertical, solid black lines represent the number of mutations connecting mitochondrial lineages. The circle sizes are proportional to the haplotype frequency as shown on the scale.

**Figure 5 insects-14-00360-f005:**
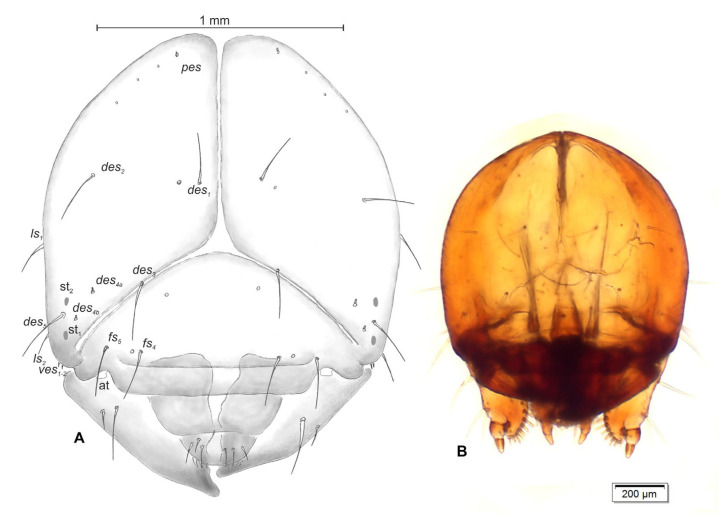
*Otiorhynchus smreczynskii,* mature larva, head. (**A**) Scheme, (**B**) photograph (at—antenna, st—stemmata. Setae: *des*—dorsal epicranial, *fs*—frontal, *les*—lateral epicranial, *pes*—postepicranial, *ves*—ventral).

**Figure 6 insects-14-00360-f006:**
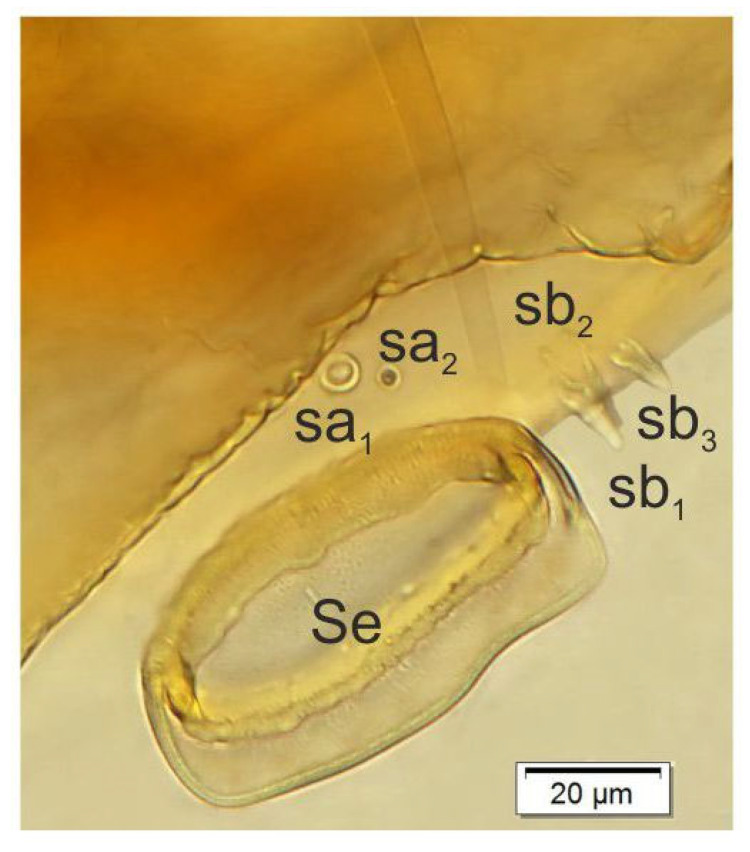
*Otiorhynchus smreczynskii,* mature larva, body parts. Right antenna (Se—sensorium, sa—sensillum ampullaceum, sb—sensillum basiconicum).

**Figure 7 insects-14-00360-f007:**
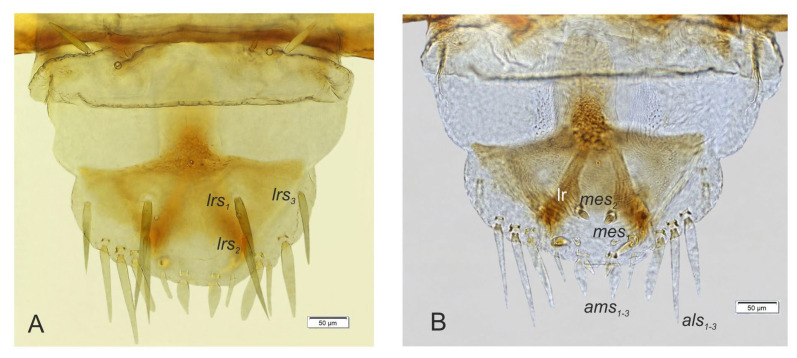
*Otiorhynchus smreczynskii,* mature larva, body parts. (**A**) Clypeus and labrum; (**B**) epipharynx (clss—clypeal sensorium, lr—labral rods. Setae: *als*—anterolateral, *ams*—anteromedial, *cls*—clypeal, *lrs*—labral, *mes*—median).

**Figure 8 insects-14-00360-f008:**
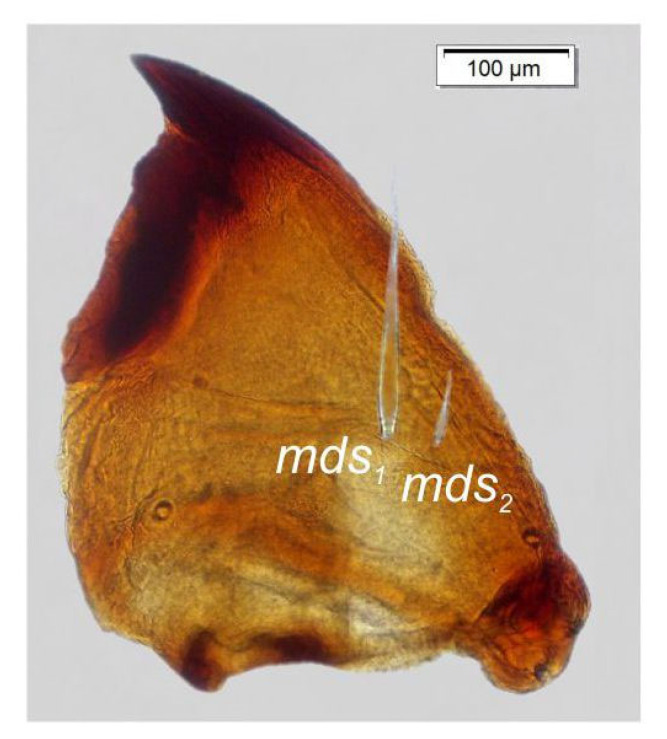
*Otiorhynchus smreczynskii,* mature larva, body parts. Right mandible (*mds*—mandibular setae).

**Figure 9 insects-14-00360-f009:**
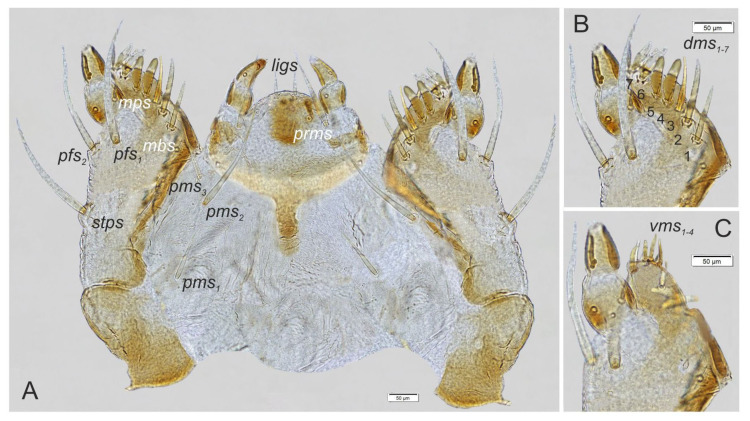
*Otiorhynchus smreczynskii*, mature larva, body parts, maxillolabial complex. (**A**) Ventral aspect, (**B**) apical part of right maxilla, dorsal aspect, (**C**) apical part of right maxilla, ventral aspect (Setae: *dms*—dorsal malar, *ligs*—ligular, *mbs*—malar basiventral, *mps*—maxillary palp, *pfs*—palpiferal, *prms*—prelabial, *pms*—postlabial, *stps*—stipal, *vms*—ventral malar).

**Figure 11 insects-14-00360-f011:**
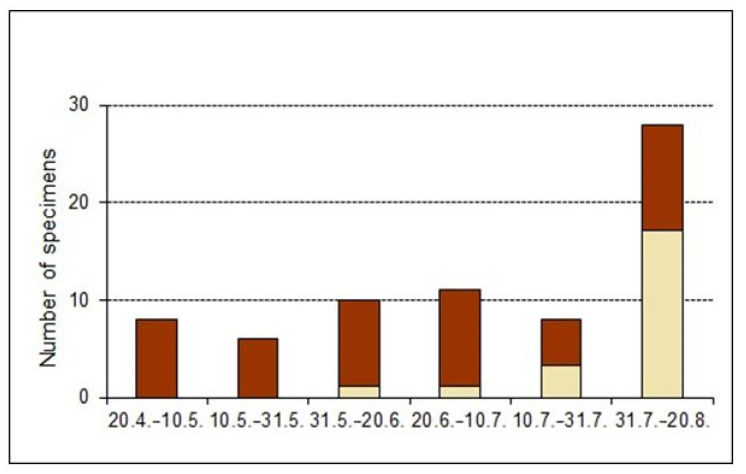
Phenology data of *Otiorhynchus smreczynskii* from pitfall trap catches in Germany, 2009 (several sites combined); proportion of light-coloured teneral adults (from [13], reworked by quantitative differentiation of teneral adults).

**Figure 12 insects-14-00360-f012:**
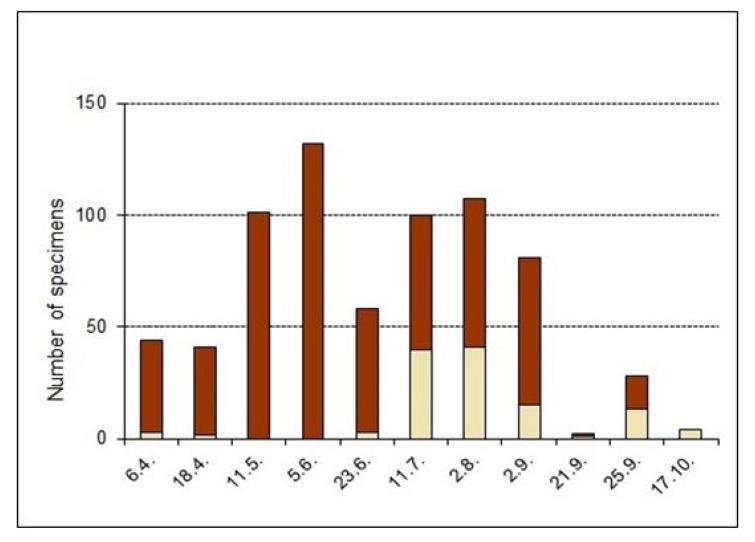
Phenology data of *Otiorhynchus smreczynskii* from sweep net catches in Hannover, 2011; proportion of light-coloured teneral adults (from [13], reworked by quantitative differentiation of teneral adults).

**Figure 13 insects-14-00360-f013:**
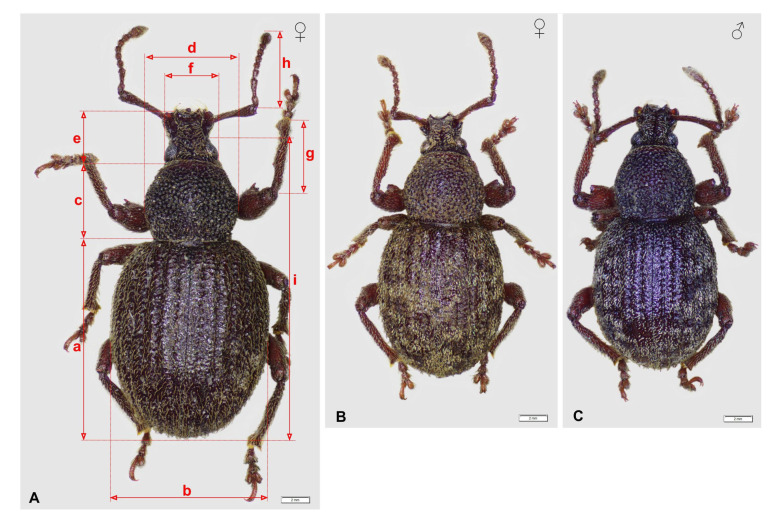
*Otiorhynchus smreczynskii.* (**A**) Female (Lublin, Poland) and *O. rotundus;* (**B**) female; (**C**) male (Gródek, Poland). a—length of elytra, b—width of elytra, c—length of pronotum, d—width of pronotum, e—length of head and rostrum, f—width of head, g—length of tibia, h—total length of funiculus and clava, i—length of the body without rostrum.

**Figure 14 insects-14-00360-f014:**
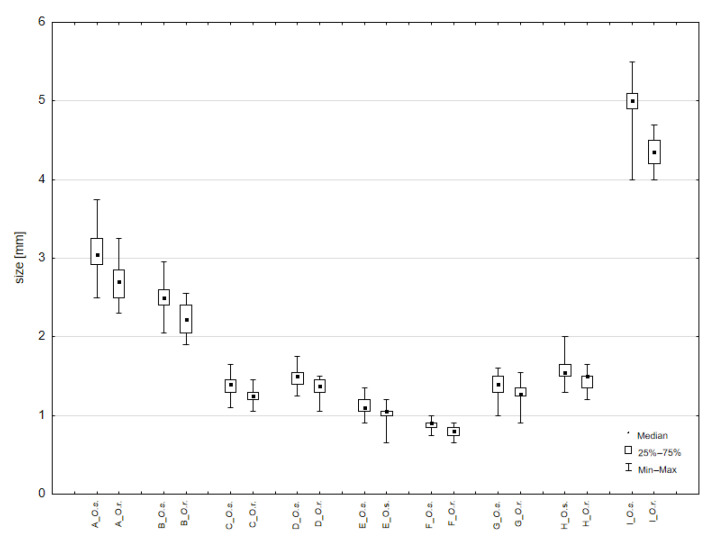
Box and whisker plot for differential variables of *O. smreczynskii* and *O. rotundus*.

**Figure 15 insects-14-00360-f015:**
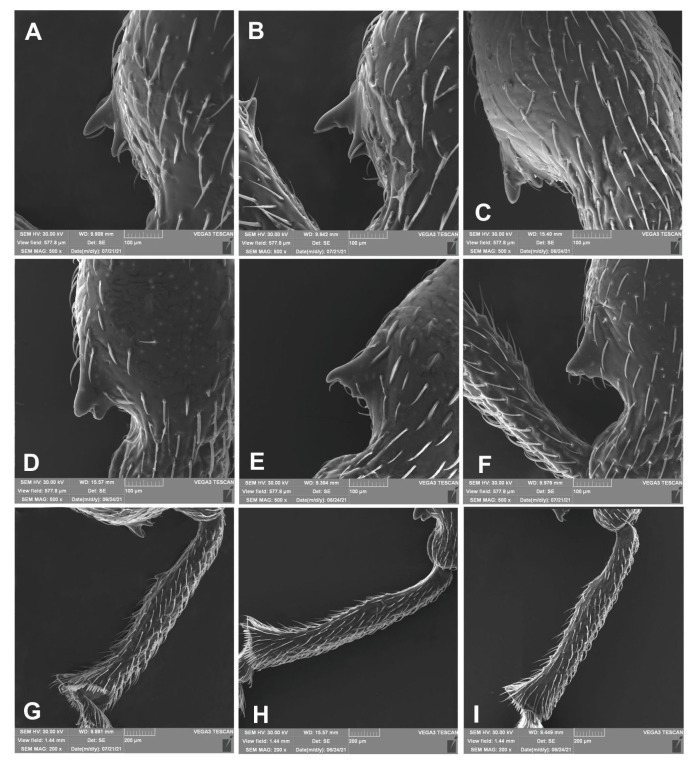
*Otiorhynchus smreczynskii* (**A**–**C**,**G**) and *O. rotundus* (**D**–**F**,**H**,**I**) (adults), variety and comparison of selected body parts. (**A**–**F**) Spine of the fore pair of legs; (**G**–**I**) spines on the inner edge of the fore legs. (**A**,**G**) Holotype, (**B**,**C**) Lublin, Poland, (**D**,**H**) male, Lublin, Poland, (**E**) female, Lublin, Poland, (**I**) female, Gródek, Poland.

**Figure 16 insects-14-00360-f016:**
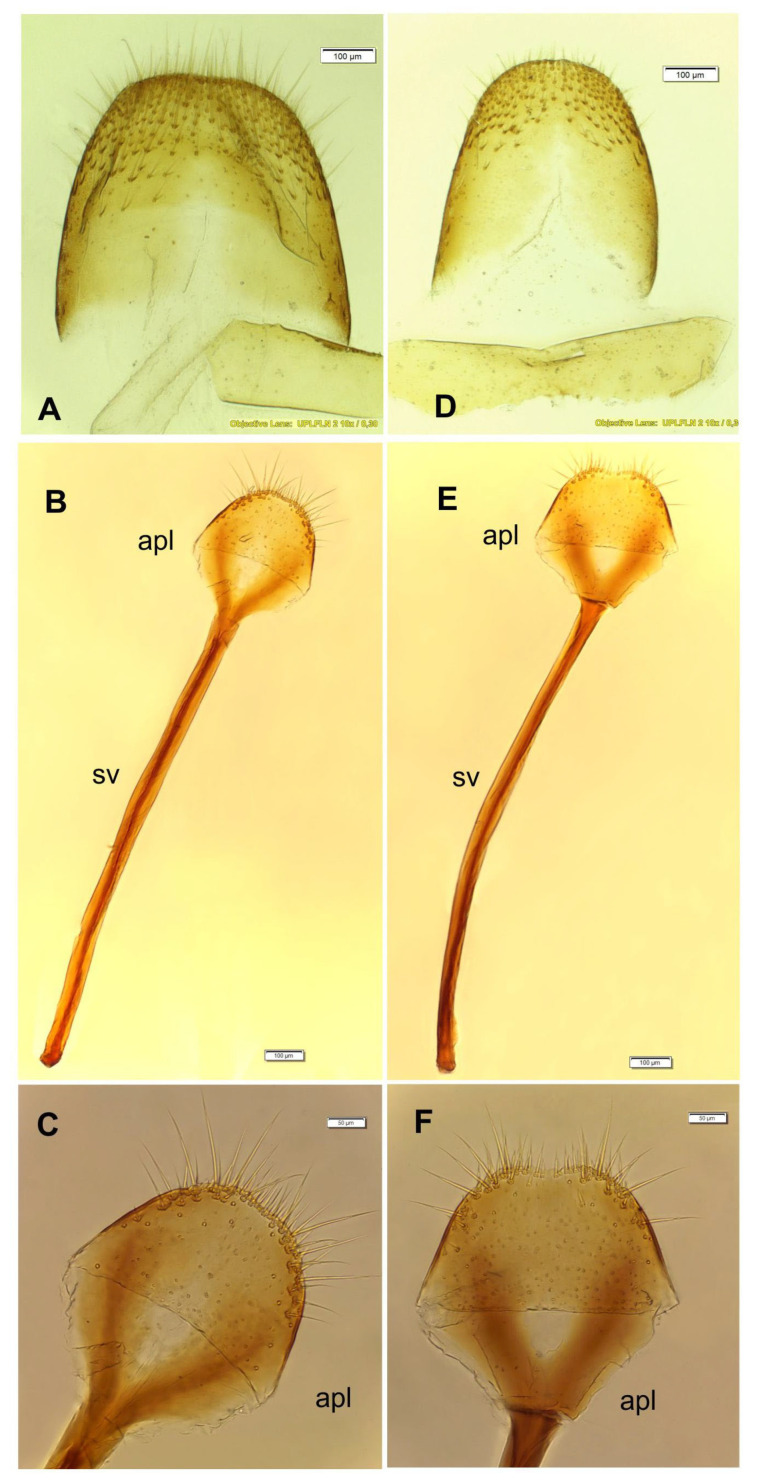
*Otiorhynchus smreczynskii* and *O. rotundus*: selected parts of female genitalia. (**A**) *O. smreczynskii*, abdominal tergite 8; (**B**) *O. smreczynskii*, abdominal sternite 8; (**C**) *O. smreczynskii*, apical lobe, magnified; (**D**) *O. rotundus*, abdominal tergite 8; (**E**) *O. rotundus*, abdominal sternite 8; (**F**) *O. rotundus*, apical lobe, magnified (apl—apical lobe, sv—spiculum ventrale).

**Figure 17 insects-14-00360-f017:**
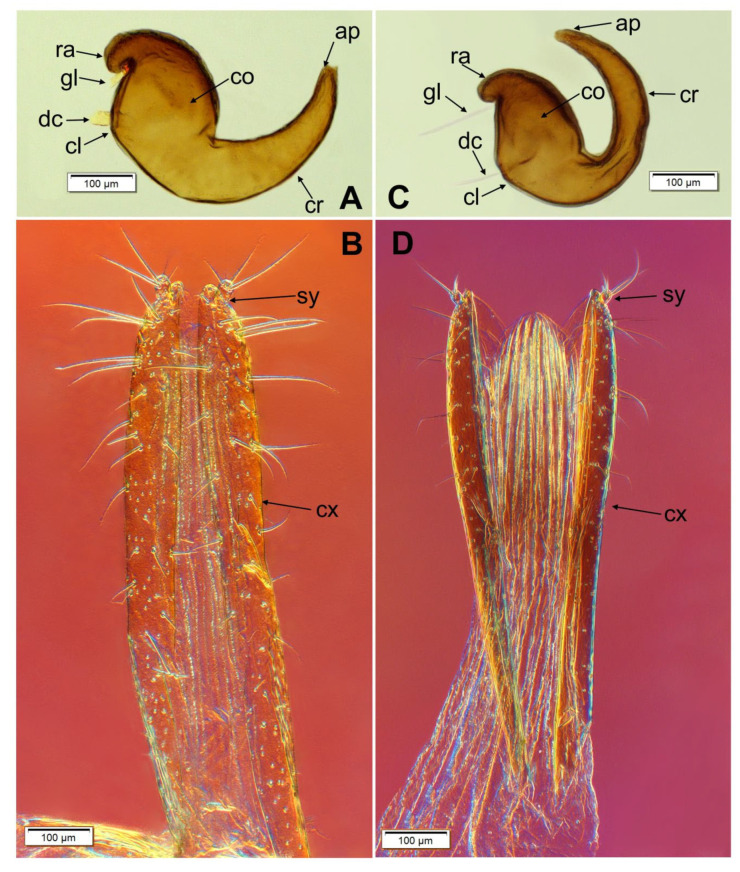
*Otiorhynchus smreczynskii* and *O. rotundus*: selected parts of female genitalia. (**A**) *O. smreczynskii*, spermatheca; (**B**) *O. smreczynskii*, ovipositor; (**C**) *O. rotundus*, spermatheca; (**D**) *O. rotundus*, ovipositor (ap—apex, cl—collum, co—corpus, cr—cornu, cx—coxites, dc—duct, gl—gland, ra—ramus, sy—stylus).

**Figure 18 insects-14-00360-f018:**
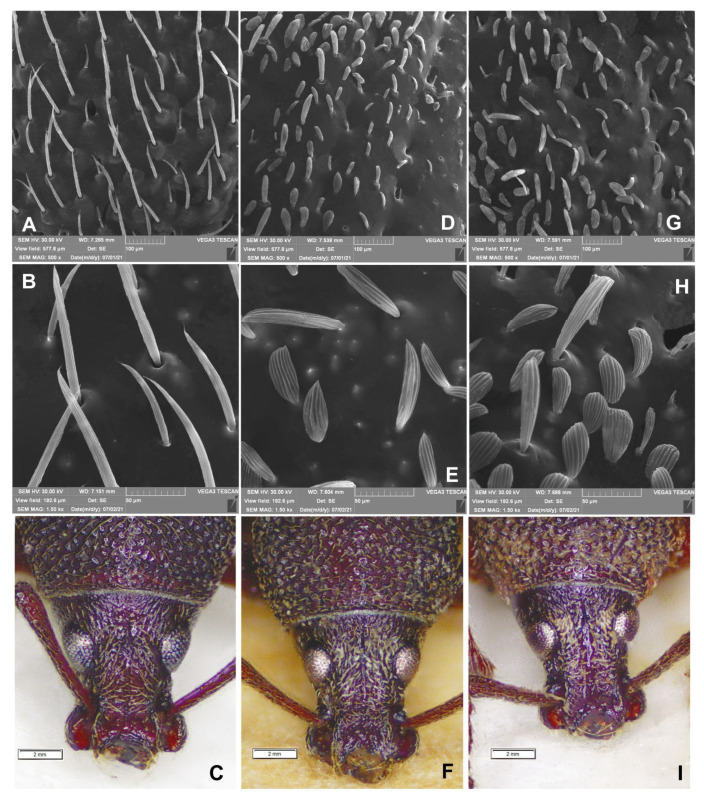
*Otiorhynchus smreczynskii* (**A**–**C**) and *O. rotundus* (**D**–**I**) (adults), variety and comparison of selected body parts. (**A**,**B**,**D**,**E**,**G**,**H**) Scales on elytra; (**C**,**F**,**I**) heads. (**A**–**C**) Lublin, Poland, (**D**–**F**) female, Gródek, Poland, (**G**–**I**) male, Lublin, Poland.

**Figure 19 insects-14-00360-f019:**
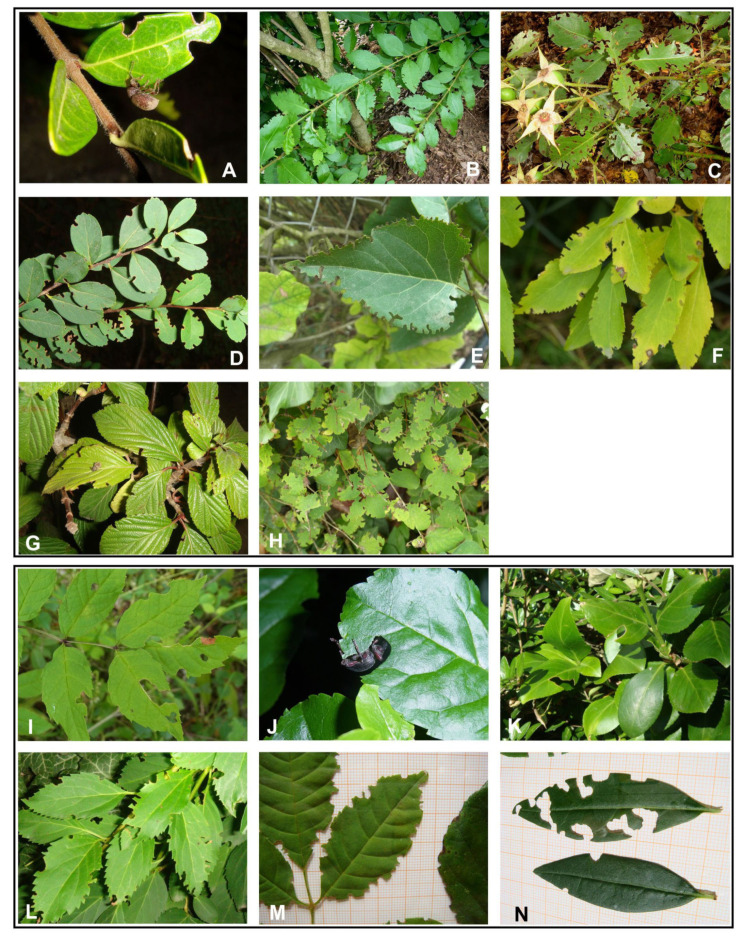
Feeding signs of *O. smreczynskii* compared with some other Entiminae species. Feeding signs of: (**A**–**H**) *O. smreczynskii*; (**I**) *O. fagi*; (**J**,**K**) *O. clavipes*; (**L**,**M**) *O. meridionalis*; (**N**) *Dodecastichus inflatus*. Plants: (**A**) *Lonicera pileata*; (**B**) *L. ovalifolium*; (**C**) *Rosa chinensis* hybrid; (**D**) *Spiraea hypericifolia*; (**E**) *Syringa vulgaris*; (**F**,**L**) *Forsythia intermedia*; (**G**) *Viburnum* × *bodnantense*; (**H**) *Symphoricarpos albus*; (**I**,**M**) *Fraxinus excelsior*; (**J**) *Euonymus fortunei*; (**K**) *Euonymus japonicus*; (**N**) *Ligustrum vulgare*.

**Figure 20 insects-14-00360-f020:**
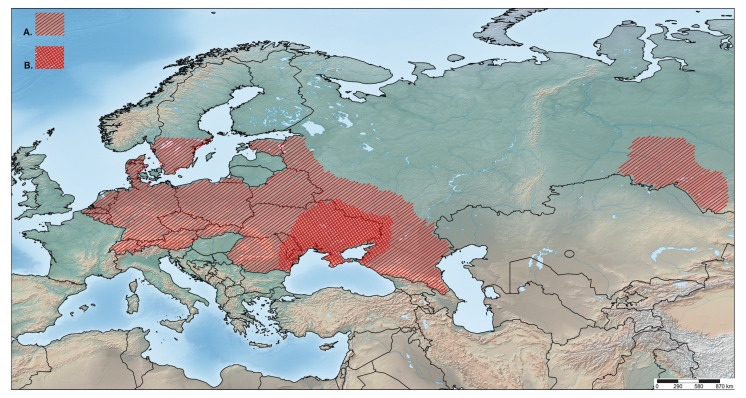
*Otiorhynchus smreczynskii*. (**A**) Distribution; (**B**) hypothetical place of origin.

**Table 1 insects-14-00360-t001:** Average mt*COI* divergence based on pairwise analysis (*p*-distance method) between all recorded haplotypes, grouped according to species affiliation. (d1) divergence over all sequence pairs; (d2) divergence over sequence pairs within groups; (P) *p*-distance over sequence pairs between groups; (S.E.) standard error.

Sequence Group	Haplotypes	d_1_ (S.E)	d_2_ (S.E)	P (S.E.)	
				1	2
1. *O. rotundus*	rot1-rot2	0.029 (0.005)	0.002 (0.001)	-	(0.008)
1. *O. smreczynskii*	smr1		n/c (n/c)	0.043	-

**Table 2 insects-14-00360-t002:** Data for instar identification in *O. smreczynskii* (data of mean value in mm).

Instar	$\mathrm{Mean}\;\mathrm{Value}\;(\overset{-}{x})$	Specimens	Source
Egg	0.435 × 0.571	11	own data
L1	0.258	7	own data
Unknown instar	0.715	2	own data
Mature larva	1.076	12	own data
Pupa	0.985	6	own data
Adult	1.015	24	own data

**Table 3 insects-14-00360-t003:** Variability of selected morphological features of *O. smreczynskii* (48 exx.) and *O. rotundus* (30 exx.) (in mm) (*: body size usually below 5.20 mm).

Feature		Species			
		*O. smreczynskii*		*O. rotundus*	
		Holotype	Others	Females	Males
a	length of elytra	3.35	2.50–3.75	2.30–3.25	2.30–2.75
b	width of elytra	2.80	2.05–2.95	2.00–2.55	1.90–2.25
c	length of pronotum	1.40	1.10–1.65	1.15–1.45	1.05–1.40
d	width of pronotum	1.70	1.25–1.75	1.15–1.50	1.05–1.45
e	length of head and rostrum	1.25	0.90–1.35	0.65–1.20	0.85–1.20
f	width of head	0.95	0.75–1.00	0.70–0.90	0.65–0.85
g	length of tibia	1.50	1.00–1.60	1.00–1.55	0.90–1.35
h	length of funiculus and clava	1.60	1.30–2.00	1.20–1.65	1.20–1.50
i	length of body	5.50	4.00–5.50 *	4.10–4.70	4.00–4.50

**Table 4 insects-14-00360-t004:** Mann–Whitney U test results (*Z*—statistical value, *p*—probability, n.s.—statistically not significant).

Case	*Z*	*p*
abdomen length (A)	5.4	<0.05
abdomen width (B)	5.64	<0.05
thorax length (C)	4.51	<0.05
thorax width (D)	3.94	<0.05
head length (E)	4.05	<0.05
head width (F)	5.07	<0.05
fore tibia length (G)	3.14	<0.05
antenna length (H)	4.39	<0.05
length of the body (I)	6.72	<0.05
A/B	1.49	n.s.
C/D	1.25	n.s.

## Data Availability

The material used for description, morphometry and comparative study is deposited in collection of Department of Zoology and Nature Protection, Maria Curie-Skłodowska University Lublin, Poland. The mt*COI* sequences obtained were deposited in the NCBI GenBank database under the following accession numbers: MZ951149-MZ951156).

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
