# Peer review of "Developmental Biology and Identification of a Garden Pest, Otiorhynchus (Podoropelmus) smreczynskii Cmoluch, 1968 (Coleoptera, Curculionidae, Entiminae), with Comments on Its Origin and Distribution"

_insects, 2023, doi:10.3390/insects14040360_

Round 1

Reviewer 1 Report

I think this is a quality paper, although some parts need to be improved. Specially the part on female genitalia must be thoroughly reviewed.

Comments on specific parts

* Line 97 The title ‘Materials’ is clearly inappropriate. There are several remarks on methodology.

* Line 101. The frequency of collecting the weevils from pitfall trap and sweeping must be indicated.

* Line 177. The three authors followed in the study of genitalia are clearly misunderstood. As an example for the female VIII sternite:

a) Wanat 2007 (reference 21). This paper deals with male terminalia. It is a mistake.

b) Franz, 2011 (reference 22). Franz (uses spiculum ventrale + lamina) and indicates that he followed, in turn, three of his earlier papers. These are the papers and the authors followed in each:

Franz, 2010a: Morphological terms usually follow ... (Howden 1995; Velázquez de Castro 1997; Wanat 2007).

Franz 2010b: Morphological terms are in accordance with ... (Howden 1995; Wanat 2007; Gaiger and Vanin 2008).

Franz, 2011: Morphological terminology is in accordance with that of ... (Howden, 1995; Velázquez de Castro, 1997; Wanat, 2007).

c) Li, K. & Liang, 2018 (reference 23)They (uses spiculum gastrale sclerotized plate) and they followed, in turn, four authors: Morphological terminology for the female genitalia of the Megalopodidae follows Snodgrass (1935), Chûjô (1952, 1953), Kasap and Crowson (1985) and Lawrence et al. (2010).

In your paper VIII sternite = (apical lobe + spiculum ventrale). Please check which author you are following to use this terminology. Because it is not in the referenced papers.

* Figure 15- cr is wrongly stated as cornu (cn)

* Line 463-469. It is not clear when the description applies to O. smreczynskii and when to O. rotundus. Please clarify.

* Line 482 figure 14C does not correspond to spermatheca.

.

Author Response

Dear Reviewer,

First of all, we would like to thank You for Your very useful and substantiated comments, with which we fully agree.

About the chapter on the description of the female genitalia. Fixed references and some issues in the photos. The entire structure has been redone. I hope this chapter is now much more accessible to readers.

About the title of the chapter "Materials". I agree that it contains information about how to collect materials. In our opinion, without them the chapter will be incomplete. We were aware of this problem, but in the end, "Materials" turned out to be the best place to put this information. Therefore, we proposed changing the title of the chapter to "Source of Materials" to indicate that the chapter contains more than just a list of materials used.

All other technical issues have been corrected as directed by the Reviewer.

Reviewer 2 Report

A very interesting article on a relatively recently described species of weevil. Distinguishing Otiorhynchus smreczynskii from O. rotundus is difficult, this paper will make it much easier to recognise both species, which is all the more important in the context of their potential harm. The authors put a lot of work into the preparation of the study, rearing of specimens, their detailed measurements, making photographs and study of biology details. In addition, genetic studies were conducted, which finaly confirmed the species distinctiveness of the both taxa. 

The paper is a complete compendium of knowledge about this interesting pair of species. 

I do not make any substantive comments on the paper, except for a few suggestions that the authors may take into account, although this is not necessary for final approval of the paper. 

Detailed comments:

- In several places the words are incorrectly moved to a new line. This is more of a comment to the editor, which will be forwarded to the editorial board,

- The authors describe the process of rearing larvae in great detail. However, they do not show it in any photos or graphics. Such an addition would greatly improve the perception of this part of the paper and would be an inspiration for other researchers wishing to undertake this difficult task of laboratory rearing of weevils larvae. 

- The authors ambiguously describe in the paper the possibility of bisexual populations of O. smreczynskii. Are the males of O. smreczynskii unknown to the authors? This is in contrast to the article by Balalaikins, M. & Bukejs, A. (2011), cited in the manuscript, where the authors provide a photo of the penis of this species. At the same time, Balalaikins & Bukejs quote information from Yunakov's 2003 article, where it is written that the species is parthenogenetic. Given the complementary nature of the authors' study, they should specifically address and comment on this misleading and ambiguous information. 

- The description of the distribution of O. rotundus is outdated. the species has already been found in Hungary and Slovakia. A fairly up-to-date distribution of the species, together with a map, is presented by Kizub & Leshchenko 2020 https://www.researchgate.net/publication/341792682_Contribution_to_the_knowledge_of_the_genus_Otiorhynchus_Germar_1822_Coleoptera_Curculionidae_fauna_of_Ukraine_Part_3

Schuh et al 2009 also reported this species from Austria. https://www.zobodat.at/pdf/KOR_79_2009_0321-0326.pdf

- line 21: is Cmol. 1968, should be Cmol., 1968

- line 410 and 460: the name of the species should be written in italics

- line 456: is spineof, should be spine of

Author Response

Dear Reviewer,

We would like to express our gratitude for your professional and helpful suggestions.

We fully agree with them. Fixed problems, added a new table with photos documenting the process of breeding weevils in laboratory conditions.

The distribution of O. rotundus has been corrected according to the information provided (thank you very much for the two references).

Finally, we had to refer to the unfortunate work of Balalaikins, M. & Bukejs, A. (2011), which evidently (adult photo and penis photo) refers to male O. rotundus.

All other technical issues have been corrected as recommended by You.

best regards

Rafal Gosik
